# SOX2 regulates acinar cell development in the salivary gland

Elaine Emmerson[1†], Alison J May[1], Sara Nathan[1], Noel Cruz-Pacheco[1],
Carlos O Lizama[2], Lenka Maliskova[3], Ann C Zovein[2], Yin Shen[3],
Marcus O Muench[4], Sarah M Knox[1*]

[1]Program in Craniofacial Biology, Department of Cell and Tissue Biology, University of California, San Francisco, San Francisco, United States; [2]Cardiovascular Research Institute, University of California, San Francisco, San Francisco, United States; [3]Institute of Human Genetics, University of California, San Francisco, San Francisco, United States; [4]Blood Systems Research Institute, San Francisco, United States

**Abstract** Acinar cells play an essential role in the secretory function of exocrine organs. Despite this requirement, how acinar cells are generated during organogenesis is unclear. Using the acini-ductal network of the developing human and murine salivary gland, we demonstrate an unexpected role for SOX2 and parasympathetic nerves in generating the acinar lineage that has broad implications for epithelial morphogenesis. Despite SOX2 being expressed by progenitors that give rise to both acinar and duct cells, genetic ablation of SOX2 results in a failure to establish acini but not ducts. Furthermore, we show that SOX2 targets acinar-specific genes and is essential for the survival of acinar but not ductal cells. Finally, we illustrate an unexpected and novel role for peripheral nerves in the creation of acini throughout development via regulation of SOX2. Thus, SOX2 is a master regulator of the acinar cell lineage essential to the establishment of a functional organ.

*For correspondence: sarah. knox@ucsf.edu

Present address: †The MRC Centre for Regenerative Medicine, The University of Edinburgh, Edinburgh, United Kingdom

**Competing interests:** The authors declare that no competing interests exist.

## Introduction

Acinar cells are essential to the function of exocrine organs including the pancreas, tracheonasal glands and salivary glands. These cells produce a mucin and/or protein-rich fluid that is transported through an interconnected ductal network to the external surface where it serves multiple roles including the protection and function of the epithelial mucosa. Although there have been extensive studies on the mechanisms of branching morphogenesis (reviewed in *Mattingly et al., 2015*), the process by which most mammalian exocrine organs develop, very little is known about the mechanisms that control acinar cell development. Moreover, the factors required for the establishment of the acinar lineage in the developing salivary gland are not known.

As for the majority of exocrine organs, salivary gland acinar and duct cells form through the expansion and differentiation of a simple cuboidal epithelium. In the submandibular salivary gland (SMG), the most characterized of the three major salivary glands (considerably less is known about the development of the sublingual (SLG) and parotid (PG)), this begins with invagination of the epithelium into the condensing mesenchyme to form a single end bud containing SOX10-positive pre-acinar cells by embryonic day (E)12 (*Lombaert et al., 2013*). Branching of the single bud into 3–5 KIT-positive end buds occurs between E12.5 and E13.0 and is followed by rapid expansion and further differentiation between E13.5-E15.0 (*Lombaert et al., 2013*; *Walker et al., 2008* and *Figure 1A*). Acinar cell differentiation can be monitored by temporal expression kinetics of genes including aquaporin (AQP) 5 at E14 (*Figure 1B*), Basic Helix-Loop-Helix Family Member A15 (MIST1) (*Pin et al., 2001*) and demilune cell and parotid protein (DCPP) at E15.0; submandibular gland

**eLife digest** The salivary glands produce fluid that contains enzymes to help us to digest our food. These glands contain a tree-like network of cells – known as acinar cells – that produce the fluid, and cells that form ducts to transport the fluid out of the glands. Both types of cells form from stem cells as animal embryos develop. Like all developing organs, the salivary glands receive many different signals that guide how they grow. However, the identity of the cues that instruct a stem cell to produce a new acinar cell or duct cell are not known.

Emmerson et al. studied how the salivary glands develop in mouse embryos. The experiments show that a protein called SOX2 – which is an essential regulator of stem cells in embryos – is required for acinar cells to form. Loss of SOX2 inhibited the production of acinar but not duct cells. Furthermore, nerves that surround the gland provide support to cells that produce SOX2 and promote the formation of acinar cells.

Further experiments suggest that the nerves also play the same role in humans. Adult organs often use developmental signals to repair or regenerate tissue. As such, understanding how an organ develops may lead to new therapies that can stimulate salivary glands and other organs to regenerate after they have been damaged in adults.

protein C (SMGc)/mucin 19 (MUC19) and parotid secretory protein (PSP) at E16.0, and MUC10 at E17.0 (*Nelson et al., 2013*). In conjunction with acinar cell development is formation of the ductal lineage, which is marked initially by KRT19 at E12.0, followed by KRT7 at E13.5 (before lumenization) and prominin-1 at E14.0 (*Nedvetsky et al., 2014*; *Walker et al., 2008*). Both acinar and duct cells are derived from a reservoir of undifferentiated basal KRT5+ progenitors that are essential to SG development (*Knox et al., 2010*). However, the mechanisms regulating specification of KRT5+ cells toward these two lineages or how the proportion of duct to acinar cells is controlled are unknown.

In addition to the known growth factor pathways that regulate epithelial branching morphogenesis (e.g. FGF10/FGFR2b [*Jaskoll et al., 2005*]), we recently discovered that neurotransmitters released from intraglandular parasympathetic nerves also control multiple aspects of SG development. The SMG and SLG receive innervation from the parasympathetic submandibular ganglion that forms at E12.0 through coalescence of neuronal cell bodies around the primary duct. The ganglion begins to innervate newly forming end buds and ducts from E12.5 (*Knox et al., 2010*), with unidirectional axon outgrowth from the ganglion toward the end buds being mediated by neurturin/GFRa2 signaling (*Knox et al., 2013*). Acetylcholine is produced as soon as the ganglion forms (*Coughlin, 1975*) and activates muscarinic receptors on the epithelium to maintain an undifferentiated stem cell population marked by KRT5 (*Knox et al., 2010*). In addition to this function, muscarinic activation also induces epithelial branching (*Knox et al., 2013*, *2010*) but whether parasympathetic nerves are specifically required for the generation of the acinar lineage is unknown.

Using the acini-ductal network of developing murine and human salivary glands in combination with in vivo and ex vivo studies, we reveal that SOX2 is specifically required to generate the acinar cells necessary for the creation of a secretory organ. We show that SOX2 is essential for the establishment and survival of acinar cells and that its expression and the proliferative expansion of SOX2-positive cells depend on neuronal acetylcholine signaling by parasympathetic nerves. As such, we identify SOX2 as a master regulator of the salivary acinar cell lineage and uncover a conserved peripheral nerve-based mechanism for selectively generating the secretory acinar cell lineage during organogenesis.

## Results

SOX2 is an important transcriptional regulator of development and maintenance in multiple organs including trachea, oesophagus and intestine (*Arnold et al., 2011*; *Gontan et al., 2008*; *Okubo et al., 2006*; *Que et al., 2009*, *2007*). In the murine E13 embryo, SOX2 is expressed by the invaginating oral epithelium (*Figure 2—figure supplement 1B*), throughout the epithelium of the SLG (*Figure 1C*) and the proximal duct of the SMG and the developing parasympathetic ganglion (*Figure 1C*; *Lombaert and Hoffman, 2010*). Genetic lineage tracing at E10.5 indicated that SOX2-

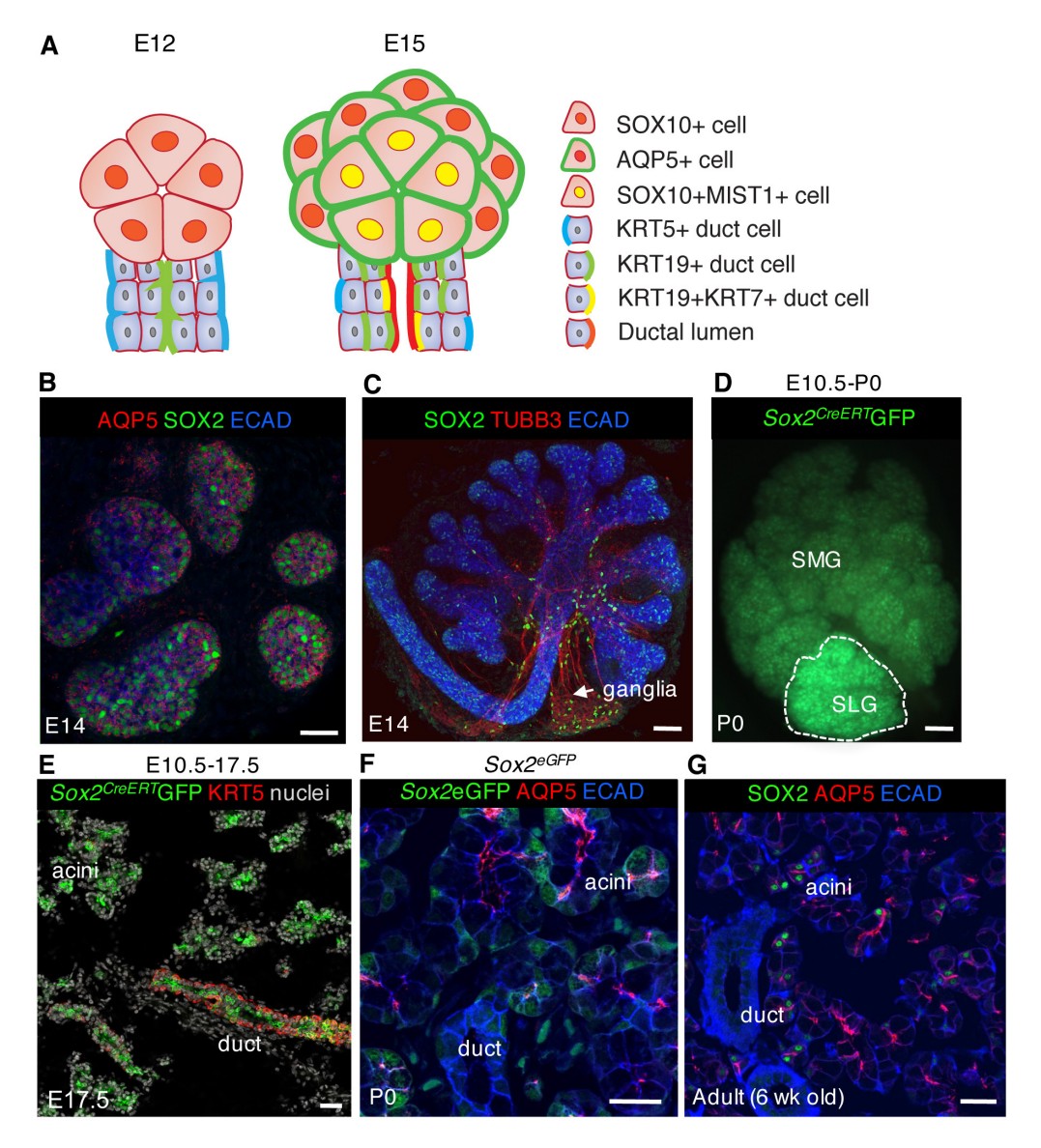

**Figure 1.** SOX2 marks a progenitor cell population in the salivary gland that gives rise to acinar and duct cells. (A) Schematic showing markers of acinar and duct cells at E12 and E15. (B) E14 SLG stained for AQP5, SOX2 and E-cadherin (ECAD). Scale bar is 20 µm. (C) Immunostaining of E14 SLG for SOX2, nerves (TUBB3) and Ecadherin-positive epithelial cells (ECAD). (D and E) Recombination was induced in Sox2$^{CreERT2}$;Rosa26$^{mTmG}$ mice at E10.5 and SMG+SLG traced until E17.5 or P0. (E) SLG were immunostained for KRT5 (red) and nuclei. (F and G) Representative images of P0 Sox2$^{eGFP}$ (F) and adult wild-type (G) SLG immunostained for the acinar marker AQP5, ECAD and/or SOX2. Scale bars in F and G are 20 µm, (C) and E are 50 µm and D is 100 µm; Images in B, E, F and G are 2 µm confocal sections and the image in C is a 30 µm projection of 4 µm sections.

positive cells are progenitors giving rise to both duct and acinar cells of the SMG and SLG (*Figure 1D,E Arnold et al., 2011*). However, we found that SOX2 becomes restricted to a subpopulation of AQP5+ acinar cells in the postnatal and adult SLG, the most poorly understood of the three major salivary glands (*Figure 1F and G*), suggesting that SOX2 might function in the expansion and/or maintenance of the acinar lineage.

To study the role of *Sox2* in salivary gland morphogenesis, a tamoxifen-inducible Cre under the control of the epithelial *Krt14* promoter was used to ablate *Sox2* prior to the onset of gland ontogenesis (E10.5; see extensive loss of SOX2 expression in epithelia but not the ganglion in *Figure 2—figure supplement 1B*). Unexpectedly, despite SOX2-positive cells giving rise to both acinar and

duct cells, genetic ablation of *Sox2* had a profound effect on the generation of acini but not ducts. The number of end buds in the SMG and SLG was significantly reduced from early stages of development and onwards, with the SLG exhibiting little to no branching (E13 to E16.5; *Figure 2A–B* and *Figure 2—figure supplement 1A–E*). Conversely, duct morphology, the number of KRT19-positive ductal cells and ductal gene transcripts *Krt7*, *Krt19* and *Egfr*, which are required for ductal development (*Häärä et al., 2009*; *Knox et al., 2010*), were similar to wild-type controls (*Figure 2A–D* and *Figure 2—figure supplement 1C*). Moreover, by E16.5, the number of SOX10-positive progenitors and differentiated AQP5-positive and MIST1-positive acinar cells were severely reduced (*Figure 2C–E*, *Figure 2—figure supplement 1C*). Further analysis of changes in gene expression profiles between wild-type control and *Sox2*-deficient SMG+SLG by RNA-seq and qPCR validation confirmed the reduction in genes associated with differentiated acinar cells (*Smgc*, *Pip*, *Spdef*, *Rab3d*; *Figure 2C*, *Figure 2—figure supplement 1G*). To rule out the phenotype being a cause of a difference in recombination efficiency between acini and ducts, we confirmed that *Sox2* ablation efficiency is comparable between AQP5+/SOX10+ acini and KRT19+ ducts (*Figures 1E* and *2F*) using a lineage-tracing model ($Krt14^{CreERT2};Sox2^{fl/fl};Rosa26^{mTmG}$).

SOX2 co-localizes with the acinar progenitor marker SOX10 in acini of developing murine glands (*Figure 2E*, left panel). Given SOX10 expression was lost upon ablation of *Sox2*, we performed lineage tracing in conjunction with *Sox2* ablation to determine if SOX10 was specifically lost in cells derived from those in which *Sox2* was ablated ($Krt14^{CreERT2};Sox2^{fl/fl};Rosa26^{mTmG}$). In this model, cells in which Cre recombinase has been activated (and consequently *Sox2* ablated) will be labeled with membrane-bound GFP+ (green), whereas cells where no recombination has occurred will retain membrane-bound Tomato (mT, red). We found that only ~5% of end bud cells deficient in *Sox2* expressed SOX10 or AQP5 in comparison to control glands ($Krt14^{CreERT};Rosa26^{mTmG}$) in which >90% express SOX10 or AQP5 (*Figure 2E and F*). We then determined whether SOX2 could directly regulate *Sox10* using ChIP-qPCR analysis of E17.5 SG. This time point was chosen due to the small size of the tissue limiting our analysis at earlier time points. We found significant enrichment of SOX2 within a specific region of the *Sox10* promoter (promoter region A; *Figure 2G*), suggesting that SOX2 may promote the generation of the acinar lineage via transcriptional control of *Sox10*. Combined, these data indicate that the maintenance of SOX10-positive acinar progenitor cells and the generation of differentiated AQP5-positive acinar cells are dependent on SOX2.

Previous studies in other organ systems show that a deficiency in SOX2 results in a cell fate switch (23–26]). However, we did not find any evidence that *Sox2*-deficient cells in the end buds divert toward a duct cell fate (*Figure 3A*, GFP+ cells do not express early ductal marker KRT19), indicating that the reduction in epithelial branching was not due to the accumulation of the ductal lineage. Therefore, we addressed if loss of end bud cells and reduced morphogenesis in *Sox2*-deficient SMG +SLG were due to apoptosis. In the absence of *Sox2*, there were activated Caspase-3-positive cells in the end buds (*Figures 2D* and *3B*) and increased expression of pro-apoptotic genes (*Bax*, *Bbc3* and *Pmaip1*; *Figure 2—figure supplement 1G*) at E16.5. Apoptosis was specific to end bud cells as no apoptosis was observed in the ducts (*Figure 3B*), suggesting a preferential requirement for SOX2 in end bud cell survival. In addition to increased apoptosis, the proliferation of end bud cells was decreased ~70% in E16.5 salivary glands lacking *Sox2* (*Figures 2D* and *3B*; Ki67+ cells). Apoptosis and reduced proliferation were not due to a loss in FGF signaling that has previously been shown to be required for epithelial survival and acinar cell expansion (*Lombaert et al., 2013*; *Matsumoto et al., 2016*) as the expression of downstream targets of FGF signaling transduction (*Etv4*, *Etv5*) and FGF ligands (*Fgf1*, *Fgf10*, *Fgf7*) were not reduced compared to wild-type controls (*Figure 2—figure supplement 1G*).

As both apoptosis and reduced proliferation were apparent at E16.5, and there was reduced branching and increased apoptotic markers at E13-E13.5 (*Figure 2—figure supplement 1A,B and F*) to further delineate between mitosis and cell death as a cause of reduced growth, we analyzed tissue at E11.5 (Cre induction at E10.5 by injection of tamoxifen). At this stage of development, the undifferentiated oral epithelium has begun to invaginate into the condensing mesenchyme. We observed an increase in apoptotic cells in the invaginating mutant epitheliums compared to wild-type controls (arrows indicate cleaved CASP3-p cells) but proliferation was not perturbed, suggesting that cell death rather than reduced proliferation impairs organ growth (*Figure 3C*). To test if inhibiting apoptosis could rescue epithelial branching that is, cell death is a causative factor in the loss of morphogenesis, we cultured E11.5 mandibles from wild-type and *Sox2*-deficient embryos ex

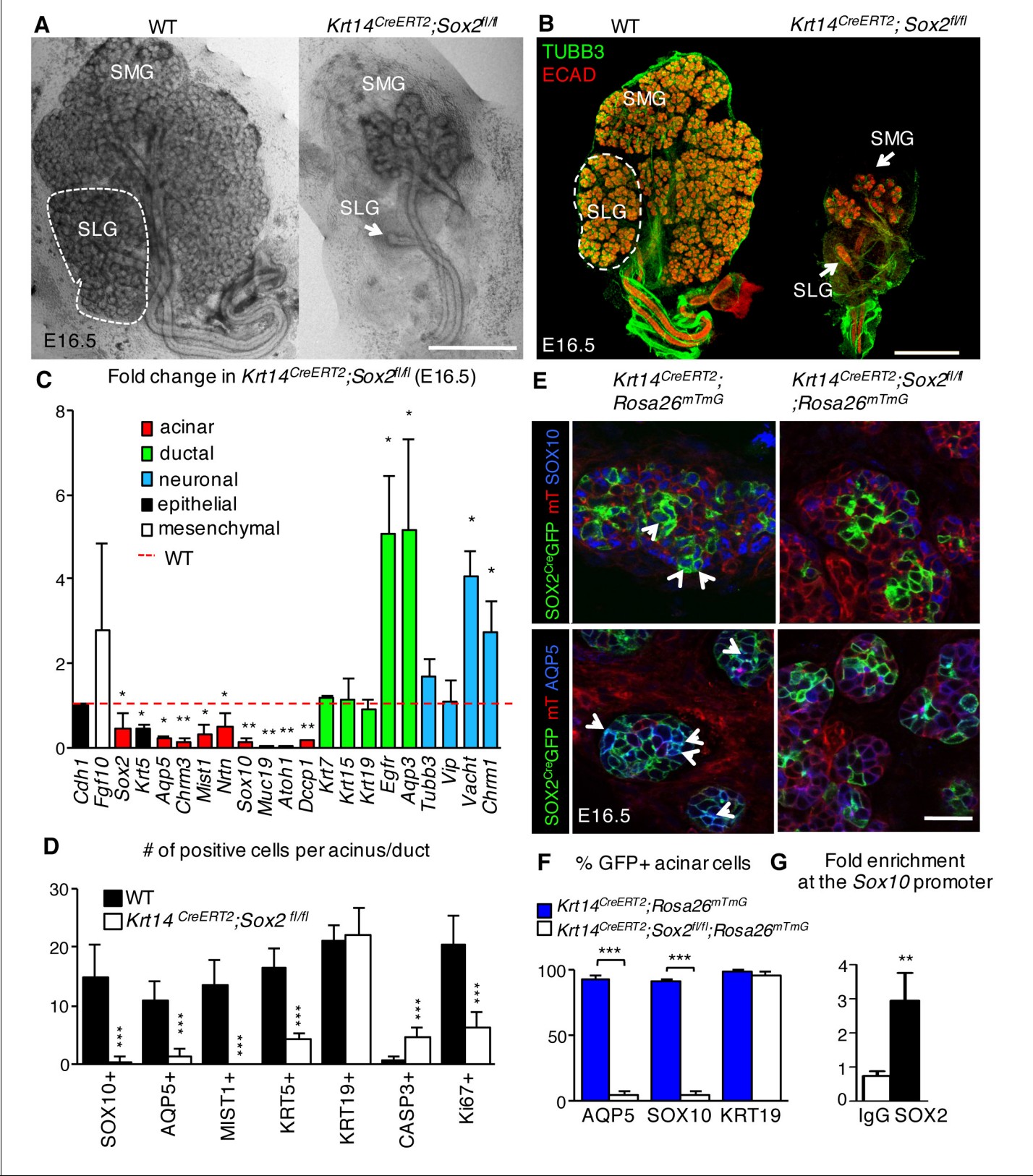

**Figure 2.** SOX2 is essential for establishing SOX10+ acini during organogenesis. (**A–D**) SMG+SLG from *Krt14^CreERT2; Sox2^fl/fl* and wild-type (WT) embryos in which recombination was induced before gland ontogenesis at E10.5 and 11.5. (**A**) Representative brightfield images of *Krt14^CreERT2; Sox2^fl/fl* and wild-type (WT) SMG+SLG at E16.5. Scale bar is 1 mm. (**B**) Representative images of SMG+SLG immunostained for nerves (TUBB3) and epithelium

*Figure 2 continued on next page*

*Figure 2 continued*

(Ecadherin, ECAD). Scale bar is 1 mm. SMG = submandibular gland, SLG = sublingual gland. Dashed white line denotes SLG. (**C**) qPCR analysis of E16.5 *Krt14*$^{CreERT2}$*; Sox2*$^{fl/fl}$ and wild-type (WT) SMG+SLG for genes involved in acinar differentiation, ductal differentiation and innervation, with expression normalized to *Rsp29*. Red dashed line = WT. n = 3 embryos per genotype. Data are means+s.d. and were analyzed using a one-way analysis of variance with post-hoc Dunnett's test. *p<0.05, **p<0.01. (**D**) Quantification of cells expressing acinar or ductal markers, cleaved caspase-3 or Ki67 in acini of E16.5 in *Krt14*$^{CreERT2}$*; Sox2*$^{fl/fl}$ and wild-type (WT). n = 2–4 glands/genotype and cells were counted in 3–4 acini/gland. Data are means+s.d. and were analyzed using a Student's *t*-test, ***p<0.001. (**E and F**) SMG+SLG from *Krt14*$^{CreERT2}$*; Rosa26*$^{mTmG}$ and *Krt14*$^{CreERT2}$*; Rosa26*$^{mTmG}$*; Sox2*$^{fl/fl}$ in which recombination was induced at E10.5 and 11.5 were immunostained for SOX10 and AQP5 (**E**) and GFP+ cells expressing SOX10 and AQP5 were quantified (**F**). n = 3 glands/genotype and cells were counted in 3–4 acini/gland. Data were subjected to a Student's *t*-test, ***p<0.001. Scale bar in **E** is 20 µm. Arrowheads indicate double positive cells. (**G**) qPCR for enrichment of *Sox10* in SOX2 ChIP. n = 20 pooled SLG, average three experiments, *p<0.05. Additional data for this figure in *Figure 2—figure supplement 1*.

The following source data and figure supplements are available for figure 2:

**Source data 1.** Source data relating to *Figure 2C*.
**Source data 2.** Source data relating to *Figure 2D*.
**Source data 3.** Source data relating to *Figure 2F*.
**Source data 4.** Source data relating to *Figure 2G*.
**Figure supplement 1.** Acini are depleted in the absence of *Sox2*.
**Figure supplement 1—source data 1.** Source data relating to *Figure 2—figure supplement 1D*.
**Figure supplement 1—source data 2.** Source data relating to *Figure 2—figure supplement 1E*.
**Figure supplement 1—source data 3.** Source data relating to *Figure 2—figure supplement 1F*.
**Figure supplement 1—source data 4.** Source data relating to *Figure 2—figure supplement 1G*.

vivo (*Knosp et al., 2015*) with the pan-caspase inhibitor Z-VAD-FMK (50 µM, *Nedvetsky et al., 2014*; *Figure 3D*) for 60 hr. However, despite a significant reduction in CASP3+ cells, extensive epithelial cell survival and growth of the epithelium, we did not measure an increase in the number of acini in *Sox2*-deficient SMG+SLG cultured with the cell death inhibitor (*Figure 3D–F*). Thus, these data reveal novel, independent roles for SOX2 in regulating epithelial cell survival and generating acini.

Our previous study indicated that KRT5+ progenitors are maintained in an unspecified state by parasympathetic nerves via acetylcholine/muscarinic activation (*Knox et al., 2010*). As a subpopulation of KRT5+ cells of the developing SMG co-express SOX2 (*Lombaert et al., 2011*), we next asked if nerves also maintain SOX2+ progenitor cells by comparing SG where the ganglia had been mechanically or genetically ablated with nerve-containing controls. SMG+SLG in which the epithelium and mesenchyme was recombined without the ganglia and cultured for 48 hr (*Knox et al., 2010*) exhibited a severe reduction in the number of acini, SOX2 protein and SOX2+ cells. However, not only were SOX2+ progenitors adversely affected, the acinar lineage was also greatly depleted in the absence of innervation. In the ganglia-free SG, we found a significant reduction in SOX10-positive acinar progenitors and differentiated AQP5-positive acinar cells (*Figure 4A–C*) compared to ganglia-containing controls. Similarly, acini, SOX10+ cells, *Sox2* and acinar gene expression were reduced in SMG+SLG derived from *Phox2b* mutant embryos that are deficient in the submandibular parasympathetic ganglion and other craniofacial ganglia (*Pattyn et al., 1999*) (*Figure 4D–F*). However, absence of innervation did not impair the expression of ductal marker KRT19 or ductal gene transcripts (*Figure 4A and F*). Thus, peripheral innervation not only preserves the progenitor cell pool, as previously found, but also selectively preserves those progenitors that contribute specifically to the acinar cell lineage.

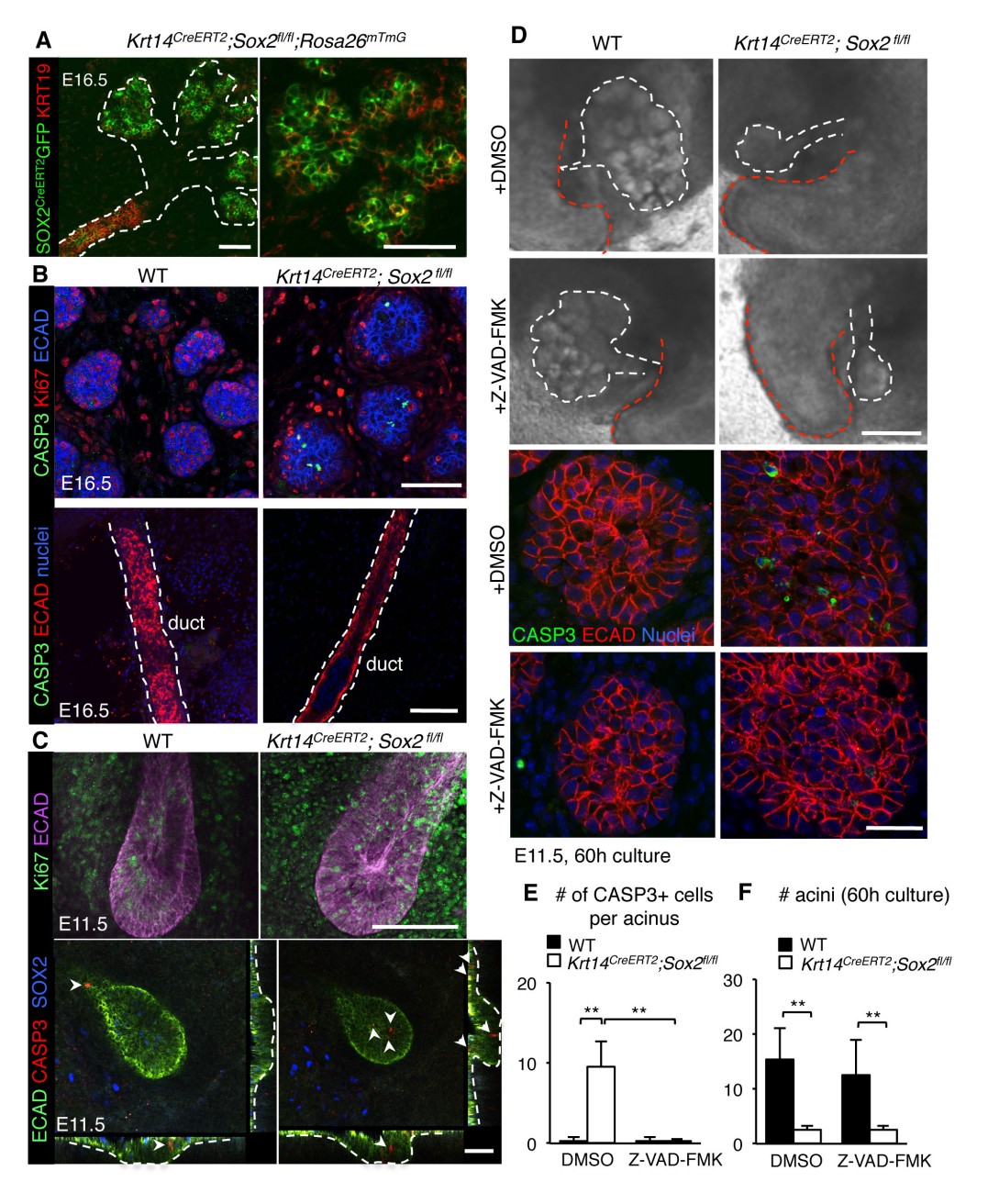

**Figure 3.** Regulation of the acinar lineage by SOX2 is independent of its function in cell survival. (A–C) Representative images of SMG+SLG from WT, *Krt14CreERT2; Sox2fl/fl; Rosa26mTmG* or *Krt14CreERT2; Sox2fl/fl* immunostained for the ductal marker KRT19, apoptotic marker CASP3, proliferation marker Ki67, SOX2, ECAD and/or nuclei. Scale bars are 50 µm. Images in A, B and lower panel C are 1–2 µm confocal sections; images in C upper panel are 20 µm projections of 1.5 µm sections. Recombination was induced at E10.5 and E11.5 for images in A and B, and E10.5 for images in C. (D) Representative images of E11.5 mandibles from *Krt14CreERT2; Sox2fl/fl* and wild-type (WT) cultured for 60 hr in the presence of DMSO or the pan-caspase inhibitor ZVAD-FMK (50 µM). SMG+SLG (lower panels) were immunostained for Ecadherin (ECAD), nuclei and cleaved caspase-3 (CASP3). White dashed line (upper panels) denotes SMGs branching off the tongue. Red dashed line indicates tongue. Scale bars are 200 µm (upper panels) and 25 µm (lower panels). (E, F) Quantification of the number of CASP3+ cells (E) and acini (F). n = 3 glands per treatment and cells were counted in 3–4 end acini per gland (E). Data in E and F are means+s.d. of three biological replicates and two experiments. Data were analyzed using a Students t-test. *p<0.05, **p<0.01.

The following source data is available for figure 3:

**Source data 1.** Source data relating to *Figure 3E*.

*Figure 3 continued on next page*

*Figure 3 continued*

**Source data 2.** Source data relating to *Figure 3F*.

Next, we determined if acetylcholine/muscarinic signaling regulates SOX2 and production of acinar cells by specifically targeting muscarinic receptors. FACS analysis showed that 94% of the SOX2-positive epithelial cells in the SLG at E15.0 express the muscarinic receptor CHRM1 (*Figure 5—figure supplement 1A*), indicating that CHRM1 activation is a potential regulator of SOX2 cells. CHRM1 is also expressed throughout the epithelium including ~42% of SOX2-negative epithelial cells (*Figure 5—figure supplement 1A and B*), indicating that this receptor is not exclusive to SOX2 + cells. In support of a direct role for muscarinic receptors in expanding SOX2+ cells, treatment of isolated E14.0 SLG epithelial rudiments devoid of nerves and mesenchyme with the muscarinic receptor agonist carbachol (CCh) for 48 hr increased the number of SOX2+ cells as well as proliferating SOX2+EdU+ cells compared to the control (*Figure 5A–B*). Furthermore, culture of mouse E14.0 SLG with the muscarinic antagonist 4-DAMP for 4 or 24 hr reduced SOX2 expression (gene and protein), the number of SOX2+ cells and SOX2+ cell proliferation (*Figure 5C–D* and *Figure 5—figure supplement 1C*. Similar to the outcomes for SMG+SLG in which *Sox2* was ablated or nerves removed, inhibition of CHRM1 decreased end bud cell proliferation, differentiated AQP5-positive acinar cells and SOX10-positive acinar progenitors (*Figure 5E* and *Figure 5—figure supplement 1D and E*), whereas KRT19-positive cells (*Figure 5E* and *Figure 5—figure supplement 1E*) or ductal lineage transcript levels were unchanged (*Krt7*) or increased (*Krt19* and *Egfr*; *Figure 5—figure supplement 1C*). In addition, there was decreased expression of acinar differentiation markers (*Aqp5*, *Mist1* and *Chrm3*; *Figure 5—figure supplement 1C*). Finally, to determine if muscarinic activation was sufficient to restore the acinar lineage after denervation, we treated nerve-free SMG+SLG explants (containing mesenchyme) with CCh for 48 hr and compared to untreated ganglia-containing and ganglia-free controls. As shown in *Figure 5F–H*, CCh was sufficient to return *Sox2* expression in the SLG, and restore AQP5+ cells, and transcript levels of acinar-specific genes including *Sox10*, *Mist1* and *Aqp5* to control levels in the SMG and SLG.

As muscarinic receptors in the salivary gland have been reported to signal via intracellular calcium and EGFR (*Jiménez et al., 2001*; *Knox et al., 2010*), we determined if either of these downstream targets was required for SOX2 and the acinar lineage. Culture of E14.0 SLG for 4 hr with the CaMK-II inhibitor KN-93 but not the EGFR inhibitor PD168393 decreased expression levels of *Sox2* and *Sox10* (*Figure 5—figure supplement 1F*). In comparison, inhibition of EGFR or calcium signaling reduced expression of more differentiated acinar markers (*Mist1, Aqp5*) suggesting that early and late markers of salivary gland development are differentially regulated. In contrast, ductal genes were expressed at similar or higher levels than controls upon KN-93 treatment (*Figure 5—figure supplement 1F*). Thus, CHRM1 and calcium signaling regulate epithelial SOX2 in the developing SLG and are required for generation of the acinar lineage.

Finally, we investigated whether neuronal signaling was an evolutionarily conserved mechanism for generating acini and preserving SOX2+ progenitors by analyzing the impact of nerves and muscarinic activation on SOX2 and the acinar lineage in human tissue. Although the murine and human glands differ in gross morphology, they undergo similar stages of morphogenesis (*Teshima et al., 2011*) and are highly innervated (*Figure 6A* and *Figure 6—figure supplement 1A*). Due to the paucity of expression data on human fetal salivary gland, we first characterized expression of acinar and duct markers as well as the location of CHRM1, SOX10 and SOX2. As shown in *Figure 6A* and *Figure 6—figure supplement 1A*, CHRM1 protein and proliferating SOX2+ and SOX10+cells are enriched in the acinar compartment of all major human fetal salivary glands, including the parotid gland (PG). We confirmed that human SMG/SLG/PG acinar cells are also enriched in SOX10, CHRM1 and MIST1 as well as CD44 and AQP3 protein (acini do not express AQP5 protein at these stages) and that the duct cells express KRT7 in addition to EGFR, KRT5, KIT and KRT14 (*Figure 6A* and *Figure 6—figure supplement 1A*). However, KRT19 was expressed in both acini and ducts, indicating that it is not a bona fide ductal marker in fetal human tissue (*Figure 6—figure supplement 1A*). To determine if parasympathetic nerves promote acinar cell formation in developing human gland, we developed a co-culture assay in which human fetal SLG explants were mixed with murine ganglia or

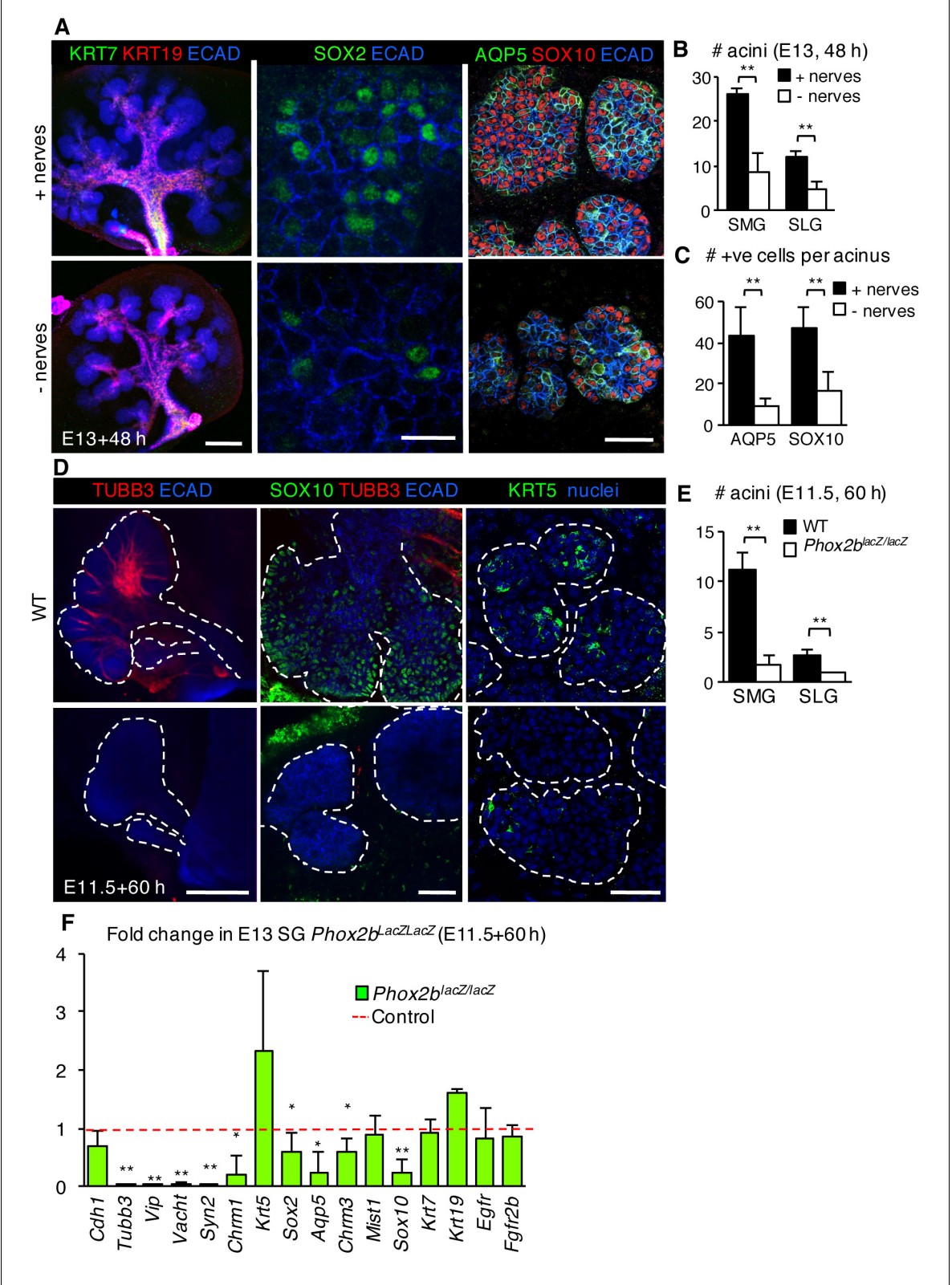

**Figure 4.** The acinar cell lineage and SOX2 are selectively depleted in the absence of parasympathetic nerves. (**A–C**) E13 murine SMG+SLG cultured for 48 hr ± parasympathetic ganglion (nerves) and subjected to immunofluorescent analysis (**A**). Glands were immunostained for markers of duct cells (KRT19, KRT7), SOX2, SOX10+ acinar progenitors, AQP5+ acinar cells and epithelial cells (E-cadherin; ECAD). The number of acini (**B**) and AQP5+ and SOX10+ cells (**C**) were quantified. (**D–F**) E11.5 murine SMG+SLG deficient in *Phox2b* were cultured for 60 hr. Glands were were immunostained for

*Figure 4 continued on next page*

*Figure 4 continued*

markers of nerves (TUBB3), acinar progenitors (SOX10), basal epithelial progenitors (KRT5) and epithelial cells (E-cadherin; ECAD; D), the number of acini were quantified (E) or qPCR was performed. (F; n = 3 embryos per genotype). Scale bars = 200 μm in (A), left panels and (D), left panels; 50 μm in (A), right panels and (D), middle and right panels; 20 μm in (A), middle panels. Data in B, C and E are means ± s.d of three biological replicates and three experiments and were subjected a Student's *t*-test. *p<0.05 **p<0.01. Data in F are means+s.d. and were analyzed using a one-way analysis of variance with post-hoc Dunnett's test. *p<0.05, **p<0.01.

The following source data is available for figure 4:

**Source data 1.** Source data relating to *Figure 4B*.
**Source data 2.** Source data relating to *Figure 4C*.
**Source data 3.** Source data relating to *Figure 4E*.
**Source data 4.** Source data relating to *Figure 4F*.

mesenchyme (*Figure 6B*). Unlike mice, the innervating human ganglion resides outside the organ and the tissue is thus denervated upon dissection. Co-culture of human SLG (20–22 w) with E13.0 parasympathetic ganglia for 7 days resulted in extensive remodeling and innervation of the epithelium (*Figure 6B and C*). As for the murine SLG, innervation increased expression of SOX2 protein (*Figure 6D*) as well as of acinar markers *MIST1* (~5–7 fold), *AQP3* (~1.6–1.9 fold), *CHRM1* (~2–10 fold) and *CHRM3* (~2.5–3.4 fold) compared to mesenchyme alone (*Figure 6E*; n = 2 fetuses). Consistent with the hypothesis that parasympathetic nerves preferentially regulate the acinar lineage, levels of ductal gene transcripts *KRT5, KRT7, KRT14, KIT* and *EGFR* were similar to human SLG co-cultured with mesenchyme only (*Figure 6E*).

To determine the impact of CHRM1 stimulation on human SOX2+ cells and the acinar lineage, we cultured human fetal SLG explants or dissociated salivary cells with CCh for 48–72 hr. As for the murine SLG, CCh treatment was sufficient to increase the expression of *SOX2* and acinar cell markers in human explants as well as dissociated cells (consisting of epithelium and mesenchyme; 20–22 weeks (w) of gestation) (*Figure 6F–G*). Furthermore, CCh treatment promoted SOX2-positive cell proliferation (*Figure 6H* and *Figure 6—figure supplement 1B*). CCh also increased expression of *KRT5* which is enriched in basal duct cells of human tissue and myoepithelial cells, suggesting these cell types are regulated by CHRM1 signaling (*Figure 6F–G*). Thus, muscarinic stimulation is sufficient to rescue the acinar cell lineage and SOX2 expression and to promote the proliferation of SOX2-positive cells in the developing human SLG. Combined, these data support a conserved role for cholinergic innervation in the generation of the acinar lineage from mice to humans.

## Discussion

Our study demonstrates a critical role for SOX2 in the generation of the acinar lineage during SG development. SOX2 acts as a master regulator of the acinar lineage by modulating the expression of lineage-specific genes and controlling both cell proliferation and survival. However, SOX2 function in acinar cell production is dependent on parasympathetic nerves that maintain its expression via acetylcholine-muscarinic-calcium signaling. The observation of similar outcomes in the human SG-nerve co-culture system provides strong evidence that the regulation of SOX2 and acini by innervation is an evolutionarily conserved mechanism that regulates the generation of acinar cells.

SOX2 also regulates cell lineage commitment in other organs such as the pituitary and skin (*Andoniadou et al., 2013*; *Driskell et al., 2009*; *Goldsmith et al., 2016*; *Lesko et al., 2013*). In the pituitary, loss of SOX2 expression in progenitors leads to a switch in cell fate between the two closely related cell types, melanotrophs and corticotrophs, in the intermediate lobe of the pituitary (*Goldsmith et al., 2016*). In the adult skin, SOX2 is expressed in dermal papilla cells that give rise to the three hair follicle subtypes and ablation of *Sox2* alters the proportion of these subtypes (*Driskell et al., 2009*; *Lesko et al., 2013*). Despite SOX2+ progenitors contributing to acini and ducts in the SMG+SLG, we unexpectedly found that *Sox2* is only required for the specification of the SOX10-positive acinar lineage. However, in contrast to the cell fate switch observed in the pituitary

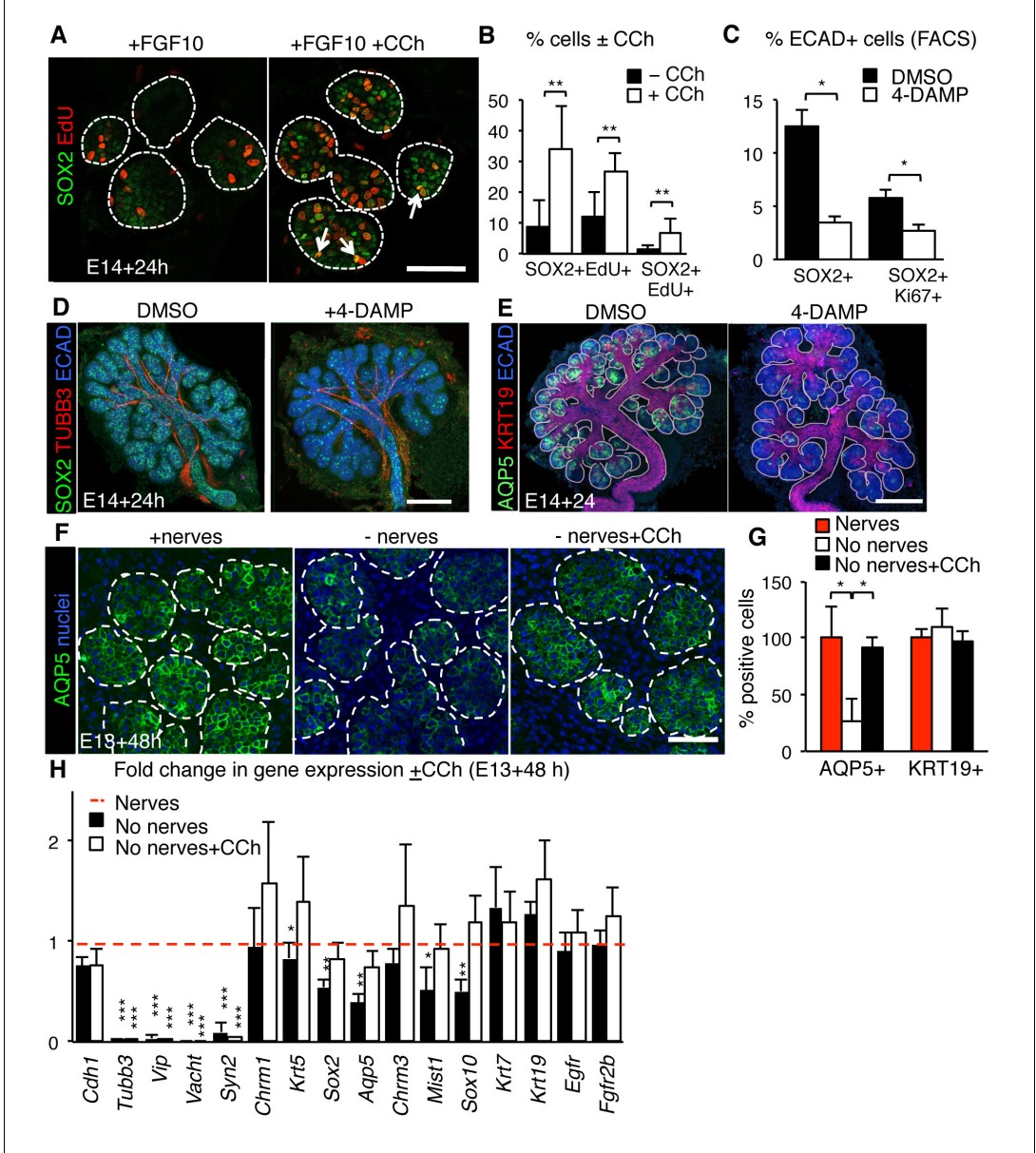

**Figure 5.** Muscarinic signaling regulates the acinar lineage and SOX2+ cells. (A–B) E14 mouse SLG epithelia cultured with FGF10 ±CCh for 24 hr. White dashed line outlines acini. Arrows indicate double positive SOX2+EdU+ cells. Scale bar is 50 μm. The number of SOX2+, EdU+ and SOX2+EdU+ cells was quantified in B. (C–E) E14 mouse SLG cultured for 24 hr with DMSO or 4-DAMP (10 μM). The number of SOX2+ and SOX2+Ki67+ cells were counted via FACS, normalized to control and expressed as percentage of total ECAD+ cells (C). In D and E, cultured glands were immunostained for SOX2, nerves (TUBB3), AQP5, KRT19 and Ecadherin (ECAD). Images are 50 μm projections of 5 μm confocal sections. Scale bars are 100 μm. (F–H) E13 SMG+SLG were cultured ± ganglia and ± CCh (100 nM) for 48 hr and immunostained for AQP5 (F) and the number of AQP5+ and KRT19+ cells counted (G). SMG+SLG were also subjected to qPCR analysis (H). Data in B, C, G and H are means ±s.d of three biological replicates and three experiments. Data in B, C and G were subjected a Student's *t*-test. *p<0.05 **p<0.01. Data in H were analyzed using a one-way analysis of variance with a post-hoc Dunnett's test. *p<0.05, **p<0.01. Additional data for this figure in *Figure 5—figure supplement 1*.

The following source data and figure supplements are available for figure 5:

**Source data 1.** Source data relating to *Figure 5B*.
**Source data 2.** Source data relating to *Figure 5C*.
**Source data 3.** Source data relating to *Figure 5G*.

*Figure 5 continued on next page*

*Figure 5 continued*

**Source data 4.** Source data relating to *Figure 5H*.

**Figure supplement 1.** Muscarinic signaling regulates the acinar lineage and SOX2+ cells.

**Figure supplement 1—source data 1.** Source data relating to *Figure 5—figure supplement 1C*.

**Figure supplement 1—source data 2.** Source data relating to *Figure 5—figure supplement 1E*.

**Figure supplement 1—source data 3.** Source data relating to *Figure 5—figure supplement 1F*.

and skin, *Sox2*-deficient cells in the end buds do not divert to ductal cell fates. This suggests that acquisition of a duct cell fate requires active re-specification and that ductal identity is not the default cell fate in the absence of SOX2 or innervation. This contrasts with a previous proposal that progenitor cells, in the absence of induction, simply differentiate into duct cells (*Knox et al., 2010*). This conclusion is corroborated by recent genetic lineage-tracing studies in the adult SG showing that acinar cells give rise to more acinar cells but not duct cells (*Aure et al., 2015*). Thus, despite SOX2 being expressed by some duct cells (*Lombaert et al., 2011*) and SOX2-positive cells being able to contribute to ducts during development, SOX2 functions in cell specification are limited to the acinar lineage.

How salivary acinar cells are established in the SG is not known. However, acinar cell expansion in the developing SG has previously been shown to be regulated by FGF10/FGFR2b signaling (*Lombaert et al., 2013*; *Matsumoto et al., 2016*). This indicates that FGF10/FGFR2b signaling is either adversely affected by loss of SOX2 or that SOX2 acts downstream of this pathway. As we did not observe a loss of FGF signaling in *Sox2*-deficient SMG+SLG and FGF10/FGFR2b signaling reduces SOX2 expression in isolated SMG epithelia (*Lombaert et al., 2011*), we conclude that SOX2 acts downstream of FGF signaling. As such, FGF signaling most likely regulates the differentiation of SOX2-positive acinar progenitors and/or their proliferative expansion.

SOX2 has been shown to promote the survival of cancer cells in vivo and has been linked to lens cell survival in Astyanax surface fish (*Boumahdi et al., 2014*; *Chou et al., 2013*; *Herreros-Villanueva et al., 2013*; *Ma et al., 2014*), but a role for SOX2 in cell survival during mammalian organogenesis has not been reported. In this study, we reveal that genetic ablation of *Sox2* but not denervation or muscarinic inhibition, which reduces but not eliminate SOX2, is sufficient to trigger cell death in the SMG+SLG. This is likely due to the levels of SOX2 remaining in epithelial cells from denervated or muscarinic receptor antagonized salivary glands being conducive to cell survival, although single-cell expression analysis is required to confirm such a mechanism. Levels of SOX2 may also influence lineage outcomes. It is possible that SOX2 may be activated in formerly SOX2-negative cells in response to cholinergic stimulation thereby specifying the acinar lineage. However, as both acinar and duct cells express SOX2, there must be other regulators at play besides SOX2 that specify this lineage. Furthermore, despite both duct and acinar cells expressing SOX2 during early gland development, apoptosis was restricted to cells in the end buds, which points to an essential and selective role of SOX2 in promoting survival of the acinar but not ductal lineage. A direct role for SOX2 in cell survival is supported by chromatin precipitation studies in skin tumor cells where a number of survival genes were directly regulated by SOX2 (*Boumahdi et al., 2014*).

We, and others, have previously shown that peripheral nerves maintain progenitors in an undifferentiated state, providing a pool of unspecified cells for developmental or regenerative processes (*Knox et al., 2010*; *Xiao et al., 2015*). We have also shown that nerves, through release of vasoactive intestinal peptide, regulate duct formation (*Nedvetsky et al., 2014*). Our current study demonstrates an unexpected role for peripheral innervation in furnishing the ductal system with acini to form a functional organ. Multiple signaling pathways such as those of the WNT and EGFR families have been reported to control epithelial differentiation and consequently tissue structure in a diverse range of organs. However, compared to neuronal signals that continuously operate during all stages of salivary gland (and organism) development, these cell/tissue intrinsic pathways tend to be

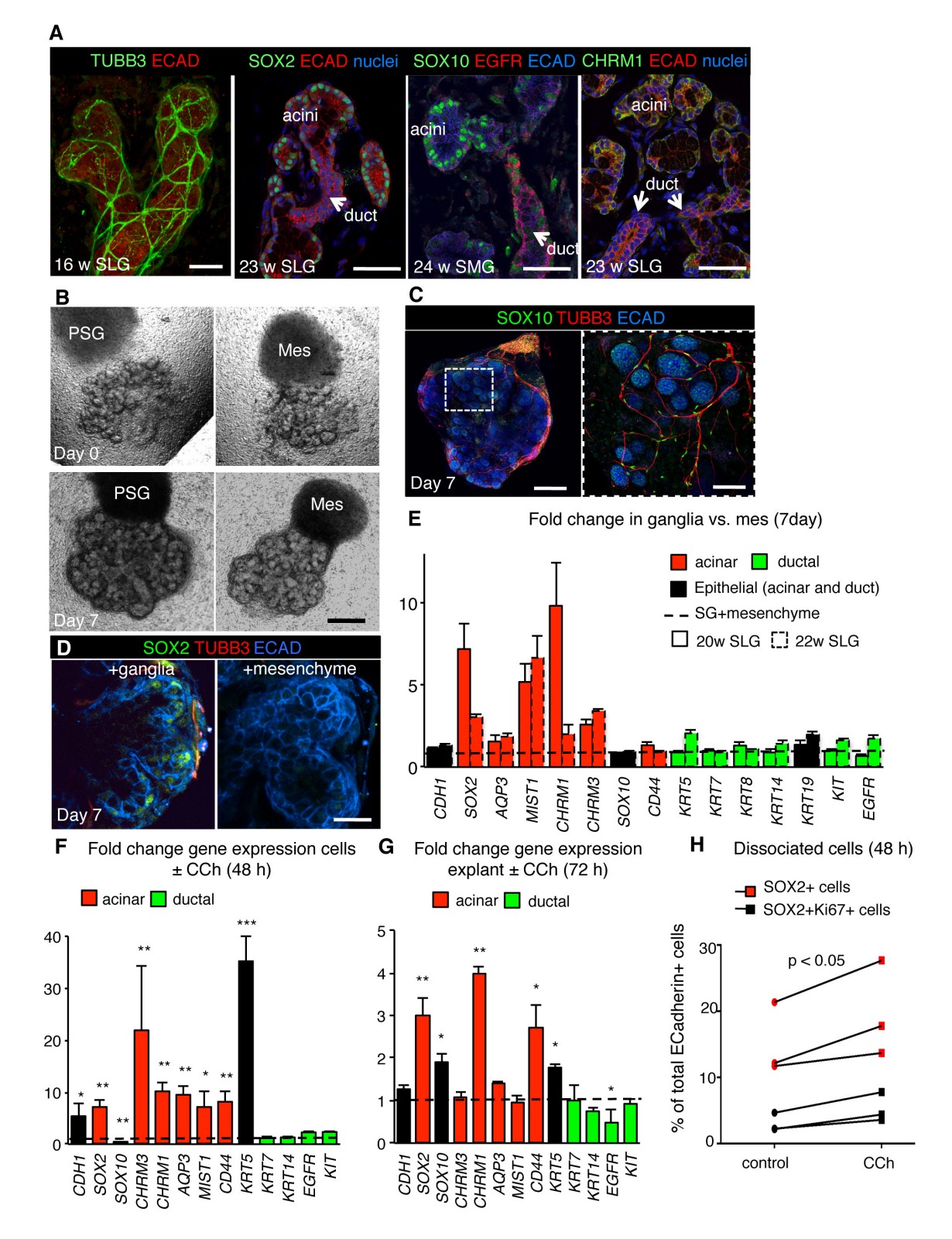

**Figure 6.** Parasympathetic regulation of SOX2 and the acinar lineage is conserved from mice to humans. (**A**) Immunofluorescent analysis of TUBB3+ nerves as well as SOX2-, SOX10-, CHRM1- and EGFR-expressing cells in fetal human submandibular (SMG) or sublingual (SLG) at 16–24 w. E-cadherin (ECAD) marks epithelial cells. Image of the 16 w SLG is a 20 μm stack of 1 μm confocal sections. All other images are 1–2 μm confocal sections. (**B**) Brightfield images of SLG explants cultured with E13 murine parasympathetic ganglion (PSG) or mesenchyme alone (MES) at day 0 (upper panels) and

*Figure 6 continued on next page*

*Figure 6 continued*

day 7 (lower panels). (**C–E**) Explants cultured with murine E13 PSG or mesenchyme for 7 days were subjected to immunofluorescent analysis for SOX10, ECAD and TUBB3+ nerves and SOX2 (**C, D**) or gene profiling by qPCR (**E**). Data in **E** (six biological replicates, two individual experiments) are means +s.d. Scale bar is 50 µm (**A**, **C** (right panel)), 500 µm (**B**, **C** (left panel), 20 µm (**D**). (**F–H**) Analysis of fetal human SLG (22–23 w) dissociated cells (**F and H**) or explants (**G**) cultured ± CCh for 48–72 hr. (**H**) The number of ECAD+SOX2+ (red markers) and ECAD+SOX2+Ki67+ (black markers) cells were measured by FACS as a percentage of cells of total ECAD+ cells. Each line represents an independent experiment. In **F** and **G** fold changes in gene expression in dissociated cells or explants were determined via qPCR with expression normalised to *GAPDH* and control values (dashed line). Data in **H** were analyzed using a Wilcoxon signed-rank test. Data in **F** and **G** were analyzed using a one-way analysis of variance with post-hoc Dunnett's test. *p<0.05, **p<0.01. Additional data for this figure in *Figure 6—figure supplement 1*.

The following source data and figure supplement are available for figure 6:

**Source data 1.** Source data relating to *Figure 6E*.
**Source data 2.** Source data relating to *Figure 6F*.
**Source data 3.** Source data relating to *Figure 6G*.
**Source data 4.** Source data relating to *Figure 6H*.
**Figure supplement 1.** SOX2 and CHRM1 are enriched in acinar cells of human fetal SG, consistent with CCh increasing proliferation of human SG SOX2+ cells.

transient in appearance, for example, FGF10 regulates acinar cell proliferation but is only expressed during early branching morphogenesis (*Steinberg et al., 2005*). As such, peripheral nerves offer a way of shaping tissue architecture throughout the development of the organism, thereby providing continuity of signals during tissue morphogenesis.

In summary, our findings support a critical role for SOX2 and nerves in establishing the secretory end units of the salivary gland, thereby selectively generating the tissue architecture necessary for organ function. Given the majority of mammalian organs receive innervation from parasympathetic ganglia, and loss of functional innervation alters epithelia and/or epithelial stem cells in multiple organs including those that express SOX2 such as taste buds, prostate and seminar vesicles, stomach and cornea (*Suzuki, 2008*; *Tatsuta et al., 1985*; *Ueno et al., 2012*; *Wanigasekara et al., 2004*; *Zhao et al., 2014*), our results may have significant implications for gaining insight into the molecular mechanisms underlying lineage specification and architectural outcomes in diverse systems.

## Materials and methods

### Mouse lines

All procedures were approved by the UCSF Institutional Animal Care and Use Committee (IACUC). Mouse alleles used in this study were provided by The Jackson Laboratory and include: $Phox2b^{LacZ/LacZ}$ (RRID:MGI:2172761) (*Pattyn et al., 1999*), $Krt14^{CreERT2}$ (RRID:MGI:5435569) (*Vasioukhin et al., 1999*), $Sox2^{CreERT2}$ (RRID:MIG:5512893) (*Andoniadou et al., 2013*), $Sox2^{fl/fl}$ (RRID:MIG:4366456) (*Smith et al., 2009*) and $Rosa26^{mTmG}$ (RRID:MIG:3623345) (*Muzumdar et al., 2007*). Sample size was calculated using the Resource Calculator (*Mead, 1988*).

### Animal experiments

#### Genetic ablation in fetal mice

$Phox2b$ homozygotes ($Phox2b^{LacZ/LacZ}$) and wild-type littermates were generated by breeding $Phox2b-LacZ$(Ki/+) female mice with a $Phox2b-LacZ$ (Ki/+) male mice (*Pattyn et al., 1999*). $Krt14^{CreER}$;$Sox2^{fl/fl}$ mice were generated from $Krt14^{CreERT2}$ (*Vasioukhin et al., 1999*) and $Sox2^{fl/fl}$ alleles (*Smith et al., 2009*). For generation of embryos deficient in $Sox2$, $Krt14^{CreERT2/+}$;$Sox2^{fl/+}$ males were crossed to $Sox2^{fl/fl}$ females and timed pregnant females injected with 5.0 mg/20 g tamoxifen (Sigma Aldrich) or corn oil (Sigma Aldrich) as vehicle at E10.5 and E11.5 and euthanized at E13, E13.5 or E16.5. $Krt14^{CreERT2}$; $Rosa26^{mTmG}$ and $Krt14^{CreERT2}$;$Sox2^{fl/fl}$;$Rosa26^{mTmG}$ embryos

were generated by crossing $Krt14^{CreERT2/+}$;$Rosa26^{mTmG}$ or $Krt14^{CreERT2/+}$;$Sox2^{fl/fl}$ males, respectively, to $Rosa26^{mTmG}$ or $Sox2^{fl/fl}$ ;$Rosa26^{mTmG}$ females and timed pregnant females injected with 5.0 mg of tamoxifen (Sigma Aldrich) at E10.5 and E11.5 and euthanized at E16.5. Pregnant female mice were euthanized by rising $CO_2$ inhalation, in line with IACUC regulations. Successful mating for all timed pregnancies was assessed by a vaginal plug and designated embryonic day 0 (E0).

## Organ culture experiments

### Sublingual gland (SLG) and mandible organ culture

In all murine ex vivo culture experiments, embryos were derived from CD1 female mice at E12-E14 (Harlan Laboratories) unless otherwise stated. SMG+SLG explants were dissected and cultured as previously described (*Steinberg et al., 2005*). SLG explants were cultured as for SMG+SLG explants after being mechanically dissected from the SMG. Mandible slices were isolated from E11.5 embryos and cultured as previously described (*Knosp et al., 2015*) for 60 hr in the presence or absence of 50 µM Z-VAD-FMK (R and D systems). SLGs were cultured with 10 µM 4-DAMP (Tocris Bioscience), 20 µM PD 168393 (Calbiochem), 15 µM KN-93 (Calbiochem) or vehicle (DMSO or water). Explants/mandibles were lysed for subsequent RNA isolation, fixed for immunostaining after 4–48 hr (4% PFA or 1:1 AcMeOH) or collected for flow cytometry.

### SLG epithelial rudiment culture

Epithelia and mesenchyme were separated using dispase treatment and mechanical dissection and cultured in a drop of laminin on a nucleopore filter over serum-free DMEM/F12 containing holotransferrin and ascorbic acid (complete media) as described for the E13 submandibular gland (*Steinberg et al., 2005*). Epithelia were cultured with 500 ng/ml FGF10 (R and D Systems) and 0.5 µl/ml heparin sulfate (Sigma Aldrich) in the presence or absence of 100 nM carbachol (Sigma Aldrich) or vehicle ($H_2O$), treated for 1 hr with EdU reagent (Click-iT EdU Alexa-Fluor 488 kit) and were fixed for immunostaining after 24–48 hr.

### Parasympathetic ganglion (PS) recombination assay

SGs dissected from CD1 timed pregnant females were dissected into epithelia, mesenchyme and PS and epithelia recombined with mesenchyme with or without the PS, as previously described (*Knox et al., 2010*). Explants were cultured for 48 hr before being lysed for RNA isolation or fixed for immunofluorescent analysis (4% PFA or 1:1 ice cold acetone: methanol (AcMeOH)).

### Human fetal SG tissue isolation and culture

Human fetal salivary glands were harvested from post-mortem fetuses between 15 and 24 weeks gestation with patient consent and permission from the ethical committee of the University of California San Francisco. Tissue was identified by location and glandular appearance and following dissection was immediately placed in 4% PFA (Electron Microscopy Sciences), RNA*later* (Qiagen) or DMEM (ThermoFisher) for live tissue explant/cell culture.

We utilized two separate assays for defining the impact of CCh on SOX2 and the acinar cell lineage: human explants and dissociated cell cultures. The explant cultures maintain tissue structure and are similar to the murine studies. Dissociated cells enabled us to define changes in gene expression of SOX2 and the epithelial lineages, as well as to perform FACS analysis for quantifying SOX2+ and SOX2+Ki67+ cells. The impact of CCh on SOX2 and acinar and ductal lineages was analyzed in both assay types.

For SG explant culture, tissue was dissected into <1 mm pieces and cultured in serum-free media (as for the embryonic mouse cultures). Tissue was incubated with 100 nM CCh for 72 hr before being lysed for RNA.

For culture of dissociated cells, tissue was first processed into single cells (see Flow Cytometry protocol below) before being cultured in DMEM/F12 (ThermoFisher) containing holotransferrin, ascorbic acid and 0.5 µg/mL heparan sulfate (Sigma Aldrich) at a density of $2 \times 10^6$ cells/mL in 12 well plates. Cells were treated with FGF10 (R and D systems) and 100 nM Carbachol (Sigma Aldrich) or vehicle ($H_2O$). Cells were cultured for 72 hr and centrifuged and lysed for RNA.

For SG explant-PSG, co-cultures tissue was dissected into <1 mm pieces and cultured on a floating filter above serum-free media. E13 mouse PSGs were isolated as described above (PS

recombination assay). One PSG per explant was placed next to the human SG and cultured for 7 days and either fixed for immunofluorescent analysis or lysed for RNA.

## Tissue processing

After fixation, postnatal and adult SGs (human and mouse) were either processed for OCT or paraffin embedding. For generation of frozen sections, tissue was incubated in increasing concentrations of sucrose (25–75%) and embedded in OCT. 12 μm sections were cut using a cryostat (Leica) and stored at −20°C. Tissue for paraffin was dehydrated by incubating in increasing concentrations of ethanol and subsequently Histo-Clear (National Diagnostics) before embedding in paraffin wax (Sigma Aldrich). 7 μm sections were cut using a microtome (Leica) and stored at room temperature.

## Immunofluorescence analysis

Whole-mount SG and tissue section immunofluorescence analysis has been previously described (*Knox et al., 2010*). In brief, tissue was fixed with either ice cold acetone/methanol (1:1) for 1 min or 4% PFA for 20–30 min followed by permeabilizing with 0.1–0.3% Triton-X. Tissue was blocked overnight at 4°C with 10% Donkey Serum (Jackson Laboratories, ME), 1% BSA (Sigma Aldrich), and MOM IgG-blocking reagent (Vector Laboratories, CA) in 0.01% PBS-Tween-20. SGs were incubated with primary antibodies overnight at 4°C: goat anti-SOX2 (1:200, Neuromics, GT15098; AB_2195800); goat anti-SOX10 (1:500, Santa Cruz Biotechnology, sc-17342; AB_2195374); mouse anti-TUBB3 (clone TUJ1 at 1:400, R and D Systems, MAB1195; AB_357520); rat anti-E-cadherin (1:300, Life Technologies, 13–1900; AB_2533005); rabbit anti-EGFR (1:200, Abcam, ab52894; AB_869579); goat anti-KIT (1:200, Santa Cruz, sc-1494; AB_631032); rabbit anti-KRT5 (1:1000, Covance, PRB-160P; AB_291581); rat anti-KRT19 (1:300, troma III, DSHB; AB_2133570); rabbit anti-KRT14 (1:1000, Covance, PRB-155P; AB_292096); mouse anti-KRT7 (1:50, Covance, MMS-148S; AB_10719738); rat anti-CD44 (Biolegend, 1:200, 103001; AB_312952); rabbit anti-CHRM1 (1:300, Santa Cruz, sc-7470; AB_2079955); mouse anti-Ki67 (1:50, BD Biosciences, 550609; AB_393778); rabbit anti-Caspase3 (1:300, Cell Signaling, 9664; AB_2070042); rabbit anti-AQP3 (1:400, Lifespan Biosciences Inc., LS-B8185; AB_2661881); rabbit anti-AQP5 (1:100, Millipore, AB3559; AB_2141915); chicken anti-GFP (1:500, Aves Labs, GFP-1020; AB_10000240); peanut agglutinin (PNA; 1:200, Vector Laboratories, AS-2074; AB2336189); and rabbit anti-MIST1 (1:500, gift from Stephen Konieczny, Purdue University). Antibodies were detected using Cy2-, Cy3- or Cy5-conjugated secondary Fab fragment antibodies (Jackson Laboratories) and nuclei stained using Hoescht 33342 (1:1000, Sigma Aldrich). EdU staining was performed using the Click-iT EdU Alexa-Fluor 488 kit. Fluorescence was analyzed using a Leica Sp5 confocal microscope and NIH ImageJ software.

## Branching morphogenesis, morphometric analysis and cell counts

Branching morphogenesis was measured by counting the number of acini (NIH ImageJ; Images were assessed by blinded researchers). All data were obtained using 4–5 SGs/group and each experiment repeated three times. For immunofluorescent analysis (e.g., *Figure 4D*), cells positively stained for markers were counted using ImageJ. Acinar cell size was measured using ImageJ. All data were obtained using 3–5 fields of view/group and each experiment repeated three times.

## Quantitative PCR analysis (qPCR)

RNA was isolated from whole tissue using RNAqueous Micro Kit (Ambion). Total RNA samples were DNase-treated (Ambion), prior to cDNA synthesis using SuperScript reagents (Invitrogen, CA). SYBR-green qPCR was performed using 1 ng of cDNA and primers designed using Primer3 and Beacon Designer software or found using PrimerBank (http://pga.mgh.harvard.edu/primerbank/). Primer sequences are listed in *Tables 1* and *2*. Melt-curves and primer efficiency were determined as previously described (*Hoffman et al., 2002*). Gene expression was normalized to the housekeeping gene *S29* (*Rps29*) for mouse and *GAPDH* for human and to the corresponding experimental control. Reactions were run in triplicate and experiments performed two to three times.

## RNA-sequencing

RNA was isolated from whole tissue using RNAqueous Micro Kit (Ambion) and total RNA samples were DNase-treated (Ambion). Samples were processed for sequencing using the TruSeq Stranded

**Table 1.** Sequences for mouse primers used for qPCR.

| Gene targ | Forward primer sequence | Reverse primer sequence |
|---|---|---|
| Aqp3 | CTGCCCGTGACTTTGGACCTC | CGAAGACACCAGCGATGGAACC |
| Aqp5 | TCTACTTCTACTTGCTTTTCCCCTCCTC | CGATGGTCTTCTTCCGCTCCTCTC |
| Ascl3 | GACAGGCTCTCGGTCTTCG | CATCTGTGTAAGAGGCCGGTA |
| Atoh1 | GAGTGGGCTGAGGTAAAAGAGT | GGTCGGTGCTATCCAGGAG |
| Bax | TGAAGACAGGGGCCTTTTTG | AATTCGCCGGAGACACTCG |
| Bbc3 | AGCAGCACTTAGAGTCGCC | CCTGGGTAAGGGGAGGAGT |
| Calm1 | TGGGAATGGTTACATCAGTGC | CGCCATCAATATCTGCTTCTCT |
| Calr | GAAGCTGTTTCCGAGTGGTTT | GCACAATCAGTGTGTATAGGTGT |
| Cnn1 | AAACAAGAGCGGAGATTTGAGC | TGTCGCAGTGTTCCATGCC |
| Ccnd1 | CATCCATGCGGAAAATCGTGG | AAGACCTCCTCTTCGCACTTC |
| Cdh1 | GACTGGAGTGCCACCACCAAAGAC | CGCCTGTGTACCCTCACCATCGG |
| Cdkn1a | CCCCCAATCGCAAGGATTCTT | CTTGGTTCGGTGGGTCTGTC |
| Chrm1 | TCCCAAGGCTCACCCAGATGTC | GCTCTGTGTGCTTTATTCTGTTGTTTCC |
| Chrm3 | CATAGCACCATCCTCAACTCTACCAAG | GGGCATTTCTCTCTACATCCATAGTCC |
| Dcpp1 | TGGTGGGGTATTATGTGGGCA | GGGATCGTTAGGGAAGCTAGA |
| Dcpp2 | ATGGGCCAATGTAGATGCTC | CCCAAGAGGCAACAGTAGGA |
| dNp63 | TTGTACCTGGAAAACAATG | GCATCGTTTCACAACCTCG |
| Etv4 | GGTCCTGTGTTCTTGGTGCTGTG | GGTCCTGTGTTCTTGGTGCTGTG |
| Etv5 | AAGCCCTTCAAAGTGATAGCGGAGAC | GTGTCCACAAACTTTCCTCTTTCTGTCAACT |
| Egfr | ACACTACGCCGCCTGCTTCAAGAG | ACTGTGCCAAATGCTCCCGAACCC |
| Fgf1 | GCACCGTGGATGGGACAAGGGACAGGAG | CACTTCGCCCGCACTTTCCGCACTGAG |
| Fgf7 | CTCTACAGGTCATGCTTCCACC | ACAGAACAGTCTTCTCACCCT |
| Fgf10 | TCTTCCTCCTCCTCGTCCTTCTCCTCTCCTTCC | CCGCTGACCTTGCCGTTCTTCTCAATCG |
| Fgfr2b | TGGCTCTGTTCAATGTGACGGAGATGGATG | AGGCGCTTGCTGTTTGGGCAGGAC |
| Kit | TGGTTGTGGTTGTTGTTGTTGTTG | GAAGGCTTGTTCCGAAGTGTAGAC |
| Krt5 | TCCTGTTGAACGCCGCTGAC | CGGAAGGACACACTGGACTGG |
| Krt7 | CGCCGCTGAGTGTGGACATCG | CTGGCTGCTCTTGGCTGACTTCTG |
| Krt8 | GGAGGAGAGCAGGCTGGAGTC | TGGTGCGGCTGAAAGTGTTGG |
| Krt14 | CCTCATCCTCTCAATTCTCCTCTGGCTCTC | CTTGGTGCGGATCTGGCGGTTGG |
| Krt15 | GCTGCTACATGCTGCTCAGGCTTAGG | CCAGGAAGGACAAGGGTCAAGTAAAGAGTG |
| Krt19 | GCCACCTACCTTGCTCGGATTG | GTCTCTGCCAGCGTGCCTTC |
| Mist1 | GCTGACCGCCACCATACTTAC | TGTGTAGAGTAGCGTTGCAGG |
| Muc19 | CTGGGTCTGGAAGTAGAAGTA | TCTAAGCCACAGAAGGAGAT |
| Nrtn | CGCTACCACACGCTGCAAGAG | TCCCACACTTATGTGAAGTCAGTTCTC |
| Pip | GGGTCTCTCATTCACATTCAGTG | TGATCTCCTGATTTTCCTGTGCT |
| Pmaip1 | GCAGAGCTACCACCTGAGTTC | CTTTTGCGACTTCCCAGGCA |
| Ptch1 | CACCCAGAAAGCAGACTACCCGAATATC | TCTCCTCCAGCATGACATACTTCACATTG |
| Prol1 | CACCTAAGCCTAGCACCTCTA | ACTTCCAAAACACTTCCGCAAAT |
| Rab3d | TACTATCGCGGAGCTATGGGT | TTTGATCTGCGTAGCCCAGTC |
| Rps29 | GGAGTCACCCACGGAAGTTCGG | GGAAGCACTGGCGGCACATG |
| Smgc | TGGCTCTGCAACACAACAGT | GGCGAAAAGCTCCCAGGTAA |
| Sox2 | CAGCATGTCCTACTCGCAGCAG | TGGAGTGGGAGGAAGAGGTAACC |
| Sox10 | ATCAGCCACGAGGTAATGTCCAAC | ACTGCCCAGCCCGTAGCC |
| Spdef | AAGGCAGCATCAGGAGCAATG | CTGTCAATGACGGGACACTG |

*Table 1 continued on next page*

Emmerson *et al*. eLife 2017;6:e26620. DOI: 10.7554/eLife.26620

*Table 1 continued*

| Gene targ | Forward primer sequence | Reverse primer sequence |
| --- | --- | --- |
| Syn2 | TAGACTGCTGTGGAGGTGAA | GCTCTGAAAGGTAAAGGTAACTG |
| Trp53 | CTCTCCCCCGCAAAAGAAAAA | CGGAACATCTCGAAGCGTTTA |
| Tubb3 | CCAGAGCCATCTAGCTACTGACACTG | AGAGCCAAGTGGACTCACATGGAG |
| Vacht | GAGTGGGAGATGGGCATGGTTTGG | GCAGGCAGGTACGACGCAAGAG |
| Vip | TCCAGTGATAGGTACTCCATCTC | CATCCATAGCACACGCAGAA |
| Zeb1 | GCTGGCAAGACAACGTGAAAG | GCCTCAGGATAAATGACGGC |
| Zeb2 | ATTGCACATCAGACTTTGAGGAA | ATAATGGCCGTGTCGCTTCG |

mRNA sample kit (Illumina) using 0.5 ug of total RNA as starting material. First strand synthesis was performed using SuperScript II Reverse Transcriptase (Thermo Fisher). Sample yield and integrity was analyzed using a Qubit (Thermo Fisher) and samples run on a 2% agarose gel. 5 nM RNA was submitted for sequencing on the Illumina platform. Reads of between 32,000 and 47,000 were obtained (n = 2 individual embryos for each genotype).

## Flow cytometry

Embryonic mouse SGs (CD1; E15.5-E16.5) or human fetal SG tissue (22–23 w) was dissected and washed in PBS containing gentamycin. Cell isolation and flow cytometry was performed as previously described (*Muench et al., 2002*). Briefly, a single-cell suspension was created by mincing tissue with a scalpel blade and incubating in a PBS solution containing Liberase TM (Roche) and DNaseI (Roche) at 37°C for 30–60 min. The enzyme reaction was quenched by the addition of FCS or BSA and the solution filtered through a 40 µm strainer (BD Falcon) and centrifuged at 1500 rpm for 5 min. The resulting cell pellet was washed with sterile PBS, centrifuged and resuspended in blocking buffer (5% serum and 0.01% NaN$_3$, Biolegend). Cell surface staining was achieved by incubating cell suspensions with antibodies against CD324 (ECAD; eBioscience, 46-3249-80; AB_1834418), CD326 (EpCAM; Miltenyi, 130-098-113; AB_2660298) and CHRM1 (Santa Cruz, sc-7470; AB_2079955).

**Table 2.** Sequences for human primers used for qPCR.

| Gene | Forward primer sequence | Reverse primer sequence |
| --- | --- | --- |
| AQP3 | GTTTCTGTGTATGTGTATGTCTGCCTTT | CGTCCCACTGCTCCTACTTATGT |
| CDH1 | AGGTGACAGAGCCTCTGGATAGA | TGGATGACACAGCGTGAG AGA |
| CD44 | CTGCCGCTTTGCAGGTGTA | CATTGTGGGCAAGGTGCTATT |
| CHRM1 | ACCTCTATACCACGTACCTG | TGAGCAGCAGATTCATGACG |
| CHRM3 | ATCGGTCTGGCTTGGGTC | CCCGGAGGCACAGTTCTC |
| EGFR | TGGCAGGTACAGTAGGATAA | CAAGTCAGTCTAACGCTCAT |
| GAPDH | CAGCCTCAAGATCATCAGCA | TGTGGTCATGAGTCCTTCCA |
| KIT | GCAGAGGAAGTGGAAGGCATCAG | TCAGTGAGACAGTAGCATTATGGAAGGT |
| KRT5 | CGTGCCGCAGTTCTATATTCT | ACTTTGGGTTCTCGTGTCAG |
| KRT7 | TCCGCGAGGTCACCATTAAC | GCTCTGTCAACTCCGTCTCAT |
| KRT8 | AAGGATGCCAACGCCAAGTT | CCGCTGGTGGTCTTCGTATG |
| KRT14 | ATCCAGAGATGTGACCTCCTC | CTCAGTTCTTGGTGCGAAGG |
| KRT19 | GTCTGCCTCCAAGGTCCTCTGA | TCTACCCAGAAGACACCCTCCAAA |
| MIST1 | CGGATGCACAAGCTAAATAACG | GCCGTCAGCGATTTGATGTAG |
| SOX2 | TGGCGAACCATCTCTGTGGT | GGAAAGTTGGGATCGAACAAAAGC |
| SOX10 | TCATCCCTTCAATGCCCCCT | TGCGTCTCAAGGTCATGGAGG |

**Table 3.** Sequences for primers used for SOX10 ChIP-qPCR.

| Primer | Forward primer sequence | Reverse primer sequence |
|---|---|---|
| A | GTGGAGGTTTGTTGATGGA | TTTGCGATGGGAGAGTCTGA |
| B | ACAGTCAGAACCTGTTGCCT | TGATACCTACTGCAGGCTGC |
| C | GCAGCCTGCAGTAGGTATCA | CTTCTTGAAGAGTAGGGC |
| D | AAAAGACAGGAACTGCCCTG | AAGGGTGCCTTCACTGAGAA |
| E | GATAGTGGGGACACAAAGAG | TCCTAATTCACTGGGCTCTG |
| F | TCTTGTTCGGGGCCTTGAAA | ATGCTTGCTGCTCCGTCCCT |
| G | AGACATCAATGAGCAGCAGG | CGCACACACACACTTTCCTA |

Subsequently, intracellular staining was achieved following fixation and permeabilization using an intracellular staining kit (eBioscience) and antibodies against SOX2 (mouse: BD Pharmingen, 562195; AB_10895118; human: R and D Systems, IC2018P; AB_357273) and Ki67 (mouse: Biolegend, 652405; AB_2561929; human: Biolegend, 350513; AB_10959326). Flow cytometry was performed on a LSRII (BD) using the appropriate single stained controls and data collected using FACSDiva (BD) and analyzed using FlowJo. 100, 000 events were collected for each sample.

## Chromatin immunoprecipitation (ChIP)-qPCR

Embryonic SGs were collected at E17 from two female CD1-ICR mice and pooled in DMEM on ice. Tissue was dissociated into a single-cell suspension as for flow cytometric analysis. ChIP was performed as previously described (*Lizama et al., 2015*). Briefly, cells were crosslinked with 1% formaldehyde, quenched with 0.125 M glycine and resuspended in lysis buffer. The chromatin was sonicated using a Biorupter sonicator (Diaganode). Rabbit anti-SOX2 antibody (Cell Signaling, #2748) or rabbit IgG control (Cell Signaling, #2729) was incubated with Pierce Protein A/G magnetic beads (Thermo Scientific) overnight at 4°C. The precipitates were washed and chromatin complexes eluted. The cross-linking was reversed (65°C for 4 hr), and the DNA was purified (QIAquick PCR Purification kit, Qiagen). 100 pg of DNA was used per PCR reaction. Primers used in PCR for quantitative ChIP are listed in *Table 3*.

## Statistical tests

Data were analyzed for statistical significance using Student's *t*-test (two groups), one-way ANOVA (multiple groups) with post-hoc testing performed using Dunnett or Tukey tests or Wilcoxon signed-rank test (GraphPad Prism or SPSS). For multiple testing, we used a false discovery rate of 0.05. All graphs show the mean +standard deviation (s.d) or mean +standard error of the mean (s.e.m).

# Acknowledgements

The authors would like to acknowledge Drs. Rushika Perera, Licia Selleri, Julie Sneddon and Rolf Zeller for their critical reading of the manuscript. We also acknowledge Wendy Fu, Minerva Loi, Shaunuk Patel, Arvin Pal and Ashley Bayer for their laboratory assistance. Funding: Funding was provided by a CIRM Postdoctoral Fellowship (EE) and NIDCR R01DE024188 (SMK).

# Additional information

## Funding

| Funder | Grant reference number | Author |
|---|---|---|
| National Institute of Dental and Craniofacial Research | R01DE024188 | Elaine Emmerson<br>Alison J May<br>Sara Nathan<br>Noel Cruz-Pacheco<br>Sarah M Knox |

California Institute for Regen-
erative Medicine

Elaine Emmerson

The funders had no role in study design, data collection and interpretation, or the decision to submit the work for publication.

## Author contributions

EE, Conceptualization, Data curation, Formal analysis, Validation, Investigation, Visualization, Meth-odology, Writing—original draft, Writing—review and editing; AJM, Data curation, Formal analysis, Validation, Investigation, Visualization, Writing—review and editing; SN, Formal analysis, Validation, Investigation; NC-P, Data curation, Formal analysis, Validation; COL, Data curation, Formal analysis, Methodology; LM, Formal analysis, Methodology; ACZ, Resources, Formal analysis, Writing—review and editing; YS, Resources, Methodology; MOM, Resources, Formal analysis, Investigation, Writ-ing—review and editing; SMK, Conceptualization, Resources, Data curation, Formal analysis, Super-vision, Funding acquisition, Validation, Investigation, Visualization, Methodology, Writing—original draft, Project administration, Writing—review and editing

## Author ORCIDs

Elaine Emmerson, http://orcid.org/0000-0002-5902-3368
Marcus O Muench, http://orcid.org/0000-0001-8946-6605
Sarah M Knox, http://orcid.org/0000-0002-7567-083X

## Ethics

Animal experimentation: This study was performed in strict accordance with the recommendations in the Guide for the Care and Use of Laboratory Animals of the National Institutes of Health. All of the animals were handled according to approved institutional animal care and use committee (IACUC) protocols (#AN107810 and AN111238) of the University of California San Francisco.

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
