## [Decision Letter]

Thank you for submitting your article "*SOX2* regulates acinar cell development in the salivary gland" for consideration by *eLife*. Your article has been reviewed by three peer reviewers, one of whom is a member of our Board of Reviewing Editors, and the evaluation has been overseen by Marianne Bronner as the Senior Editor.

The reviewers have discussed the reviews with one another and the Reviewing Editor has drafted this decision to help you prepare a revised submission.

Summary:

This manuscript provides ample and convincing evidence that *Sox2* regulates a progenitor population in the salivary gland. The authors use a combination of state-of the art genetic lineage tracing in mice, molecular analysis and genetic deletion to show that *Sox2* marks precursors for acinar cells in the gland and necessary for the development of the gland. ChIP experiments confirm that Sox10 is a potential transcriptional target of *Sox2* in these progenitor cells. The authors further link neural regulation of *Sox2* progenitors. Overall, the study has clarified that *SOX2* is required for acinar cell maintenance and expansion but not ductal cells at the stages analysed, and that this expansion depends on ACh signalling from parasympathetic ganglia.

Essential revisions:

The reviewers agreed that the authors should analyze proliferation and apoptosis in more detail.

The data suggest that apoptosis is involved in the *Sox2* null phenotype given the increase in caspase+ cells in the glands lacking Sox 2 but inhibition of apoptosis signaling does not rescue the buds. Can the decrease in acinar cell numbers be attributed only to a proliferation defect or additionally to (i) loss of acinar fate or (ii) a reduction in primordium size at earlier stages? At which point are defects first apparent? Results for proliferation at E16.5 demonstrate a clear reduction, but how soon after deletion at E10.5 is the proliferation defect detected? The survival experiments indicate that apoptosis is not responsible for this reduction in mutants, but the timing of the experiment is not clear- is there any possibility that a bout of apoptosis has already taken place shortly after deletion, prior to the effects of the ZVAD treatments? Perhaps the timing of application of the inhibitor or the dose should be re-evaluated? Perhaps a caspase 3-specific inhibitor would be more effective (M-791)?

---

## [Author Response]

*Essential revisions:*

*The reviewers agreed that the authors should analyze proliferation and apoptosis in more detail.*

*The data suggest that apoptosis is involved in the Sox2 null phenotype given the increase in caspase+ cells in the glands lacking Sox 2 but inhibition of apoptosis signaling does not rescue the buds. Can the decrease in acinar cell numbers be attributed only to a proliferation defect or additionally to (i) loss of acinar fate or (ii) a reduction in primordium size at earlier stages? At which point are defects first apparent? Results for proliferation at E16.5 demonstrate a clear reduction, but how soon after deletion at E10.5 is the proliferation defect detected? The survival experiments indicate that apoptosis is not responsible for this reduction in mutants, but the timing of the experiment is not clear- is there any possibility that a bout of apoptosis has already taken place shortly after deletion, prior to the effects of the ZVAD treatments? Perhaps the timing of application of the inhibitor or the dose should be re-evaluated? Perhaps a caspase 3-specific inhibitor would be more effective (M-791)?*

We thank the reviewers for their comments on the timing of the defect. We have performed additional experiments at earlier time points to delineate between apoptosis and proliferation as a cause of reduced primordium size. We find that apoptosis in the *Sox2*-deficient epithelium occurs as early as E11.5, 24 h after tamoxifen injection and the time at which the epithelium invaginates into the mesenchyme. However, proliferation at this stage is similar to the control. We also show that apoptotic markers, as well as reduced branching, are apparent by E13. We have added this additional data to Figure 3 and Figure 2—figure supplement 1. These new data suggest that reduced size is due to an initial round of cell death, presumably after recombination at or after E11.

We apologize that the description of the methods for the mandible cultures was not clear. We isolated mandible slices at E11.5 following recombination at E10.5 (we assume recombination takes place at E11 given the need to activate tamoxifen) and immediately cultured these mandibles with the pan-caspase inhibitor Z-VAD-FMK thereby preventing any further apoptosis. The dose of Z-VAD used here has already been optimized in the salivary gland ex vivo model (Nedvetsky et al., 2014) and we are confident in the efficiency of inhibition, as shown by the absence of cell death in Figure 3. As we found that inhibition of cell death from E11.5 does not increase the number of acini, we conclude that *SOX2* regulates both cell survival and morphogenesis.

We have added text to the manuscript to clarify the timing of the phenotype (apoptosis and proliferation is included) as well as the timing of the cell death inhibitor assay as follows:

“As both apoptosis and reduced proliferation were apparent at E16.5, and there was reduced branching and increased apoptotic markers at E13-E13.5 (Figure 2—figure supplement 1) to further delineate between mitosis and cell death as a cause of reduced growth we analysed tissue at E11.5 (Cre induction at E10.5 by injection of tamoxifen). […] Thus, these data reveal novel, independent roles for *SOX2* in regulating epithelial cell survival and generating acini.”